# Edge Contamination, Bulk Disorder, Flux Front Roughening, and Multiscaling in Type II Superconducting Thin Films

**Michel Geahel [1,2,3], Isabelle Jouanny [3], Dominique Gorse-Pomonti [3], Marie Poirier-Quinot [1], Javier Briatico [2] and Cornelis Jacominus van der Beek [3,***

[1] Laboratoire Imagerie à Résonance Magnétique Médicale Multi-Modalités, Université Paris Sud, Université Paris Saclay, 91405 Orsay, France; michel.geahel@u-psud.fr (M.G.); marie.poirier-quinot@u-psud.fr (M.P.-Q.)

[2] Unité Mixte de Physique CNRS/Thales, Université Paris-Sud, Université Paris-Saclay, 91767 Palaiseau, France; javier.briatico@thalesgroup.com

[3] Laboratoire des Solides Irradiés, Ecole Polytechnique, CNRS, CEA, Université Paris-Saclay, 91128 Palaiseau, France; isabellejouanny@hotmail.fr (I.J.); Dominique.gorse-pomonti@polytechnique.edu (D.G.-P.);

* Correspondence: kees.vanderbeek@polytechnique.edu

**Abstract:** We have investigated the effect of different types of disorder on the propagation, roughness, and scaling properties of magnetic flux fronts in a type II superconductor. A progression from the usual (Kardar–Parisi–Zhang-type) scaling to multiscaling is observed as the disorder strength is increased. A hierarchy of disorder strengths is established for $YBa_2Cu_3O_{7-\delta}$ thin films. The results cast light on the physical origin of the roughening of flux fronts, and they are of interest for the design and elimination of flux noise in microscopic superconducting thin-film devices.

**Keywords:** type II superconductivity; vortex matter; flux pinning; critical current density; flux penetration; critical state; disorder; irradiation; roughening of elastic manifolds; nonlinear diffusion

## 1. Introduction

Because of its importance in limiting sustainable currents in superconducting wires and tapes, as well as in determining electromagnetic noise in superconducting thin-film devices, nonlinear magnetic vortex diffusion in type II superconductors has been widely studied for several decades [1–6]. In particular, the temporal evolution of the vortex density and the local screening current density is thought to give clues as to the thermal depinning mechanism and, more generally, on the type of magnetic flux pinning that is at the origin of the sustainable current [7,8]. This is because thermally activated creep of the elastic vortex system through the quenched disorder potential in the superconducting material determines the shape of the superconductor's current–voltage characteristic [9,10]. The latter's nonlinearity is responsible for the mode of magnetic flux penetration peculiar to superconductors and described by the Bean model [11]. In particular, the vortex density is such that the net sustainable screening ("critical") current is constant in the magnetic flux-penetrated regions; the latter are separated from the Meissner state by a well-defined flux front, corresponding to the furthest advance of superconducting vortex lines into the superconducting material (see Figure 1b). In general, the flux front is, as the vortices themselves, "roughened"—that is, it features random excursions from its average position, as schematically illustrated in Figure 1c [12–14]. The extent of these excursions is determined by the underlying disorder, as well as by the interaction between

vortex lines and between vortex lines and the screening current. While this feature of the flux front may seem academic, the roughening of the flux front is what will determine the onset of flux noise in superconducting thin-film devices.

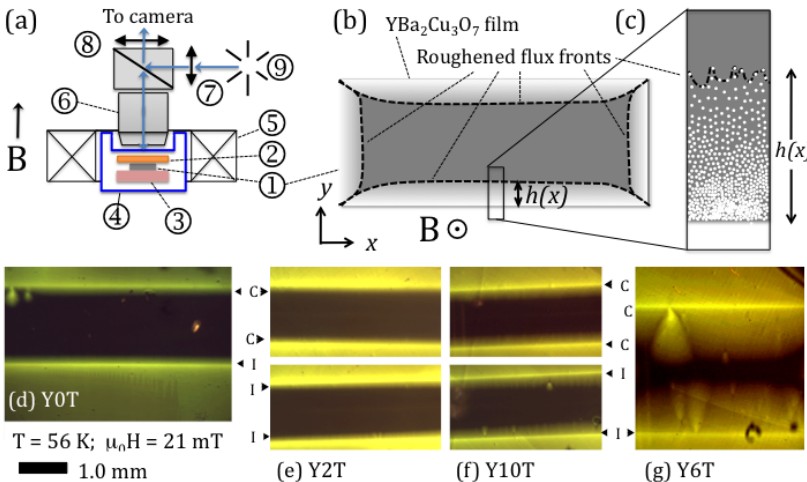

**Figure 1.** (**a**) Scheme of the experimental set-up (transverse section). The superconducting sample (1) is placed on the copper cold finger (3) and is covered with the ferrimagnetic garnet indicator film (2). The assembly is cooled down in the cryostat (4). The magnetic field is applied perpendicularly to the plane for the sample and the indicator film, using the electromagnet (5). Flux penetration is observed though magneto-optical imaging, using a polarised optical microscope. Shown are the objective (6), the polarizer (7), the analyzer (8) and the white light source (9). (**b**) Sketch of magnetic flux penetration into one of the superconducting YBa$_2$Cu$_3$O$_7$ rectangular thin-film strips (planar view with the magnetic field applied perpendicular to the plane). Magnetic flux vortices, of a diameter too small to be resolved on the scale of the sample, penetrate the superconductor from the four edges. The flux density $B$, proportional to the vortex density, is indicated by the grey scale; white indicates high $B$, and the darkest shade of grey corresponds to $B = 0$ in the Meissner state in the centre of the superconductor. The dotted lines indicate the flux fronts, or fronts of farthest vortex penetration $h(x)$. (**c**) Magnified sketch of magnetic flux penetration, indicating how a very large number of individual vortices (schematically drawn as white dots) contribute to the total magnetic flux. (**d**–**g**) Large-area magneto-optical images of magnetic flux penetration in samples (**d**) Y0T (non-irradiated reference film), (**e**) Y2T, (**f**) Y10T, and (**g**) Y6T. The latter shows a macro-defect both on the upper and the lower (long) edges. Such defects act as preferential "gates" for flux penetration and are excluded from the analysis. In all figures (**d**–**g**), the temperature is 56 K, and the applied field $\mu_0 H = 21$ mT.

The similarity of the advancing flux front to wetting phenomena [15]; domain-wall propagation [16,17]; and possible self-organised criticality in systems as diverse as forest fires [18], sand and rice piles [19], and braiding river networks [20] has prompted descriptions of the front itself in terms of nonlinear diffusion equations. Surdeanu et al. [12] first reported that, depending on the distance (or "height") $h$ that the flux front had penetrated superconducting YBa$_2$Cu$_3$O$_{7-\delta}$ thin films, its motion could be described either by the directed percolation depinning (DPD) model [21–24] or by the Kardar–Parisi–Zhang (KPZ) equation [25]:

$$\frac{\partial h}{\partial t} = \nu \nabla^2 h + \mu (\nabla h)^2 + \eta(x, h; t) + F \qquad (1)$$

In the KPZ model, diffusion is modified by the directed growth perpendicular to the interface, expressed by the term $\mu (\nabla h)^2$ in Equation (1); here, this term can be considered a consequence of the vector nature of the Lorentz force on the vortices. The correlations of the stochastic noise term $\eta(x, h; t)$ are assumed to be Gaussian, where $\langle \eta(x, h; t) \eta(x', h'; t') \rangle = 2D\delta(x - x')(h - h')\delta(t - t')$,

corresponding to uncorrelated point-like disorder. In superconductors, the noise term is thought to reflect the quenched disorder in the sample and is considered to be time-independent, so that $\eta = \eta(x, h)$. One then speaks of the quenched KPZ equation. Conversely, the inclusion of only thermal, "annealed" disorder that is at the origin of thermally activated creep is described by $\eta = \eta(t)$. Later work [26] reports the spatial and temporal fluctuations of the two-dimensional roughened "flux landscape" to be more adequately described by the Edwards–Wilkinson (EW) equation [27], which reads as Equation (1) but without the second (nonlinear) term on the right-hand side (RHS). Different model descriptions reflect different universality classes of the diffusion dynamics. As a consequence, the relevance of a particular model description to the physics underlying vortex motion and, notably, directional and dissipative terms, as well as correlations of the disorder, can be inferred from the scaling behaviour of the height–height correlation function:

$$C^2(x, t) = \left\langle \left[ \delta h(x', \tau) - \delta h(x' + x, \tau + t) \right]^2 \right\rangle_{x', \tau} \tag{2}$$

Here, $\delta h(x, t) = h(x, t) - \langle h(x, t) \rangle_x$ corresponds to the local deviation of $h$ from the average front position $\langle h(x, t) \rangle_x$, and $\langle ... \rangle_{x', \tau}$ denotes an averaging over the spatial coordinate $x'$ and time $\tau$. In particular, one expects that

$$C(x, 0) \propto x^{\alpha} \quad (x \ll l_{sat}) \tag{3}$$

$$C(0, t) \propto t^{\beta} \quad (t \ll t_{sat}) \tag{4}$$

where $l_{sat}$ and $t_{sat}$ are, respectively, a saturation length and time beyond which deformations of the flux front are independent. In the case under consideration, the temporal coordinate corresponds to the value of the applied magnetic field $H$ [26,28]. The roughness exponent $\alpha$ and growth exponent $\beta$ are characteristic of the universality class, and therefore reflect the underlying dynamics. The values of $\alpha$ and $\beta$ in different universality classes are given in Table 1.

**Table 1.** Values of the roughness exponent $\alpha$ and the growth exponent $\beta$ for different models; $d$ is the dimension of the roughened interface; (q)KPZ represents the quenched Kardar–Parisi–Zhang models, respectively; DPD represents the Directed Percolation Depinning model; and (q)EW and (a)EW are the (quenched) and (annealed) Edwards–Wilkinson models, respectively.

| Universality Class | Roughness Exponent, $\alpha$ | $\alpha(d = 1)$ | Growth Exponent, $\beta$ | $\beta(d = 1)$ |
|---|---|---|---|---|
| KPZ [23] | $\frac{1}{2}$ | $\frac{1}{2}$ | $\frac{1}{3}$ | $\frac{1}{3}$ |
| DPD [23] | 0.63 | 0.63 | 0.63 | 0.63 |
| aDPD [24] | 0.5 | 0.5 | 0.25 | 0.3 |
| qKPZ [29] | $(4 - d)/4$ | $\frac{3}{4}$ | $(4 - d)/(4 + d)$ | $\frac{3}{5}$ |
| EW [23] | $(2 - d)/2$ | $\frac{1}{2}$ | $(2 - d)/4$ | $\frac{1}{4}$ |
| qEW (2d) [26] | 0.75 | 0.75 | 0.5 | 0.5 |
| aEW [24] | 0.48 | 0.48 | 0.25 | 0.25 |
| 2D Oslo model [19] | 0.38 | 0.38 | 0.5 | 0.5 |

In spite of the applicability of the scaling relations (Equation (3); see Table 2), it was recently recognised that these do not constitute a complete description of the magnetic flux penetration process in superconductors, or of roughening processes in general [17,28,30,31]. In particular, disorder correlations [30], arising from the presence of extended defects in the superconductor bulk or along its edges, can significantly modify the noise term $\eta(x, h)$. In particular, fluctuations in vortex positions and the progression of the flux penetration front can be differently affected by disorder between different positions $x$, so that the noise term has a different functional dependence on $x$ in different regions of the superconductor. Second, the coalescence of front portions that are subject to different disorder realizations, such as those occurring in avalanche dynamics of penetrating magnetic flux or

in the presence of "rare events" such as edge or extended defects [17,32], will give rise to a multi-affine front [28]. Third, the fact that the front is determined by the superposed flux from individual vortex lines (or "particles") means that, in experiments, one observes a multifractal "hull function" rather than the front itself [28,33]. Each of these situations will give rise to non-trivial "multiscaling", in which the $q$-th moment of the correlation function

$$C_q(x,t) = \left\langle \left[ \delta h(x',\tau) - \delta h \left( x' + x, \tau + t \right) \right]^q \right\rangle_{x',\tau}^{1/q} \tag{5}$$

features scaling of $C_q(x,0) \propto x^{H_q}$, where $H_q$ is the (non-trivial) generalized Hurst exponent. Such multiscaling was observed in strongly pinning superconducting $Ba(Fe_{1-x}Co_x)_2As_2$ single crystals [28], in flux avalanches in Nb thin films [28,34,35], for domain walls in ferro-electric $Pb(Zr_{0.2}Ti_{0.8})O_3$ thin films [17], and in burning paper [30], as well as in the fracture of geological materials [36]. However, while different possible origins of multiscaling were posited by Grisolia et al. [28], its physical origin could not be determined. We note that the KPZ, DPD and EW models lead to a $q$-independent Hurst exponent; the values are recalled in Table 1.

**Table 2.** Results of the scaling analysis of the spatial and temporal correlations of penetrating magnetic flux (vortex) fronts in a variety of superconducting materials.

| Superconductor | Thickness | Substrate | Reference | $d$ | $\alpha$ | $\beta$ | Interpretation |
|---|---|---|---|---|---|---|---|
| $YBa_2Cu_3O_{7-\delta}$ | 80 nm | $NdGaO_3$ | Surdeanu et al. [12] | 1 | 0.64 | 0.65 | DPD |
| | | | | | 0.46 | | KPZ |
| $YBa_2Cu_3O_{7-\delta}$ | 80 nm | $NdGaO_3$ | Welling et al. [37] | 1 | 0.4 | 0.5 | Oslo model (rice pile) |
| $YBa_2Cu_3O_{7-\delta}$ | 80 nm | $NdGaO_3$ | Aegerter et al. [26] | 2 | 0.8 | 0.6 | qEW (2D) |
| $MgB_2$ | 400 nm | $Al_2O_3$ | Lucarelli et al. [38] | - | 0.58 | - | DPD |
| Nb | 100 nm | Si | Vlasko-Vlasov et al. [39] | - | 0.7 | - | - |
| $YBa_2Cu_3O_{7-\delta}$ | 80 nm | $NdGaO_3$ | Wijngaarden et al. [34] | 2 | 0.7 | 0.5 | From avalanche analysis |
| $Ba(Fe_{0.93}Co_{0.07})_2As_2$ | 30 μm | - | Grisolia et al. [28] | 1 | 0.5 | 0.33 | KPZ & multiscaling |
| $Bi_2Sr_2CaCu_2O_{8+\delta}$ | - | - | Barness et al. [29] | - | 0.75 | 0.6 | qKPZ |

In what follows, we investigate the effect of different types of controlled disorder on the scaling properties of flux fronts in $YBa_2Cu_3O_{7-\delta}$ thin films prepared by pulsed laser deposition. We investigate two types of edge disorder, introduced by chemical and Ar ion etching procedures, respectively, and four types of bulk disorder, corresponding to the as-grown films and films irradiated with different fluences of swift heavy ions. Different irradiation energies were used, corresponding to values of energy deposited by electronic excitations above the threshold of $S_e \sim 20$ keV/nm for the observation of amorphous columnar defects in the material [40,41]. Three doses were explored to vary the bulk defect density. The penetration of magnetic flux vortices in the films was then analyzed using the multiscaling procedure of Grisolia et al. [28]. Whereas as-grown films with edges defined by Ar ion etching reproduced the behaviour found by Surdeanu et al. [12] in the KPZ regime, the introduction of strong edge disorder by chemical etching or strong bulk disorder by irradiation induced multi-affine flux fronts and multiscaling. We discuss our findings in terms of the occurrence of rare events (defects) introduced at the boundary by the chemical etching procedure or introduced in the bulk by very large fluences of swift heavy ions.

## 2. Materials and Methods

The measurements of magnetic flux penetration were performed on four thin films of $YBa_2Cu_3O_{7-\delta}$ cut from the same wafer, with lateral dimensions of $10 \times 10$ mm$^2$ and thicknesses of 670 nm. The last of these was a $YBa_2Cu_3O_{7-\delta}$ M-type thin film, deposited on a Lanthanum Aluminate (LAO) substrate by reactive thermal evaporation, and acquired from THEVA Inc. (nowadays CERACO, Ismaning, Bavaria, Germany) [42]. Three of the films were irradiated with high-energy lead ions at GANIL (Grand Accélérateur National d'Ions Lourds) in Caen, France (see Table 3). The samples denoted by "Y2T" and "Y6T" were irradiated with 1 GeV $^{207}$Pb$^{56+}$ ions ($n = 1 \times 10^{11}$ cm$^{-2}$ and

$3 \times 10^{11}$ cm$^{-2}$, respectively), while a third film, denoted by "Y10T", was irradiated with 130 MeV $^{207}$Pb$^{31+}$ ions ($n = 5 \times 10^{11}$ cm$^{-2}$). The film denoted by "Y0T" was used as a non-irradiated reference. The irradiations resulted in the creation of continuous amorphous ion latent tracks in the YBa$_2$Cu$_3$O$_{7-\delta}$ material, as well as in the substrate. However, transmission electron microscopy (TEM) of our samples showed bombardment with 130 MeV ions to yield tracks with a diameter of less than 5 nm, while 1 GeV ions yielded track diameters of between 6 and 12 nm. Each ion impact was expected to result in one latent track, so that the track density $n$ ought to have corresponded to the ion fluence and to dose-equivalent fields $B_\phi \equiv n\Phi_0$ of 2 and 6 T for films Y2T and Y6T, respectively ($\Phi_0 = h/2e$ is the quantum of magnetic flux). The defect densities observed in our TEM study were $6.1 \times 10^{10}$ cm$^{-2}$ for film Y2T and $2.0 \times 10^{11}$ cm$^{-2}$ for Y6T. Therefore, there was a track deficiency of around 35%.

After irradiation, the samples were exposed to different etching procedures so as to define rectangular strips with dimensions of $\sim(4 \times 1)$ mm$^2$. In each case, one of the long boundaries of the strip was defined using a chemical etching procedure, while the opposite boundary was defined using Ar ion etching. Each sample was cleaned, first with acetone and then with isopropanol. The parts of the sample that were not to be etched were protected with synthetic resin S1813 [43]. The solution used for chemical etching was a mixture (available from MicroChemicals GmBH, Ulm, Bavaria, Germany) of acetic acid, nitric acid, phosphoric acid, and water specifically designed for this type of thin film , and held at a temperature of 6 °C.

To cool down our samples, we used a helium flow cryostat. The experiments were conducted at temperatures of 46, 56 and 66 K. The magnetic field (of up to 70 mT) was applied using a split-coil electromagnet.

The magneto-optical imaging technique was used to visualise the penetration of magnetic flux vortices into the YBa$_2$Cu$_3$O$_{7-\delta}$ thin-film samples (Figure 1a). This method consists of placing a ferrimagnetic garnet indicator film (with a thickness of 5 μm) with in-plane anisotropy on top of the sample [44,45]. The component of the local magnetic induction $B(r)$ perpendicular to the garnet film induces a Faraday rotation of the polarization of the light through the garnet. An Al mirror on the hind side of the garnet reflects the impinging light, which is then observed using a polarized-light microscope with nearly crossed polarizer and analyzer. Regions with a nonzero $B$ then show up as bright when observed through the analyzer; a higher intensity corresponds to a higher local magnetic flux density. Measurements were performed at a constant temperature by increasing the magnetic field from 0 to 67.2 mT in 1.5 mT steps. For the Y10T sample, different polarizer settings were used to optimise acquisitions at low (from 0 to 29.8 mT) and high field (from 28.9 to 59.7 mT) values. Comparative images of large areas for all the studied films are shown in Figure 1.

**Table 3.** Overview of the studied samples, with the energies and dose-equivalent fields of the Pb ion irradiation.

| Sample | Y0T | Y2T | Y6T | Y10T |
|---|---|---|---|---|
| Pb ion fluence (cm$^{-2}$) | 0 | $1 \times 10^{11}$ | $3 \times 10^{11}$ | $5 \times 10^{11}$ |
| Dose equivalent field $B_\phi$ (T) | 0 | 2 6 | 10 | - |
| Energy (GeV) | - | 1.0 | 1.0 | 0.130 |
| Critical temperature $T_c$ after irradiation (K) | - | 87.1 | 85.6 | 89.3 |
| Critical current density $j_c$ (46 K) (A·m$^{-2}$) | $1.35 \times 10^{11}$ | $1.2 \times 10^{11}$ | $(5.4 \pm 0.3) \times 10^{10}$ | $(9.5 \pm 0.05) \times 10^{10}$ |

The obtained images were calibrated by measuring the spatially resolved magneto-optical response of the garnet, that is, intensity versus applied magnetic field, above the critical temperature $T_c$ of the YBa$_2$Cu$_3$O$_{7-\delta}$ sample. All data was processed offline using Matlab (R2014a, The Mathworks). A pixel-by-pixel calibration and correction procedure were applied to determine the value of the magnetic flux density $B$ in each point of the images, and to correct for heterogeneities in the illumination. Considering that we had $N$ pixels per image and that we performed this calibration for $M$ different magnetic fields, a second order polynomial $I_{ij} = a_i H_j^2 + b_i H_j + c_i$ was defined, relating the intensity

$I_{ij}$ in the *i*-th of the *N* image pixels to the *j*-th applied magnetic field in the calibration sequence, $H_j$. The image intensity could then be expressed as $\mathbf{I} = \mathbf{H} \cdot \mathbf{C}$, where $\mathbf{I}$ is the image vector (of length *N*),

$$\mathbf{H} = \begin{pmatrix} H_1^2 & H_1 & 1 \\ \vdots & \vdots & \vdots \\ H_M^2 & H_M & 1 \end{pmatrix} \tag{6}$$

is defined as the ($M \times 3$) magnetic field matrix, and $\mathbf{C} = [a_i, b_i, c_i]$ is the ($3 \times N$) coefficient matrix (*i* ranges from 1 to *N*). The polynomial coefficients $a_i$, $b_i$, and $c_i$ could then be obtained through matrix inversion, where $\mathbf{C} = (\mathbf{H}^{-1} \cdot \mathbf{I})$, and the flux density *B* in each pixel could be obtained through the usual application of the quadratic formula.

The study of the progression of the magnetic flux fronts into the superconducting films was limited to those areas in which the images were not affected by extrinsic features such as extended defects in the magneto-optical indicator or the proximity of the lateral edges of the $YBa_2Cu_3O_{7-\delta}$ samples, which induce the curvature of the flux front. Additionally, in order to exclude any spurious effect due to the inhomogeneity of the light source, two 200 μm-wide strips straddling the image boundary were excluded from further analysis.

The flux fronts were extracted by collecting the flux density profiles perpendicular to the sample edge (the Bean profiles) in 5 pixel-wide strips, subtracting any spurious constant "induction" value in the film centre so that the Meissner state corresponded to $B = 0$, and searching for the *y*-coordinate at which the measured flux density first rose above a defined threshold value $B_t$ (starting from the film centre). For vanishing $B_t$, the flux front corresponded to the interface between the vortex-free Meissner state (dark upper part of the images in Figures 2, 3a,b, 5a–c and 6a–c), and the mixed state. It turns out however, that choosing too low a threshold value (typically, below 1.25 mT) is detrimental to the signal-to-noise ratio of the extracted data; the camera noise rather than vortex progression influences the features of the determined front. For all experiments, we therefore extracted the magnetic flux fronts for a series of $B_t$ values of between 1.5 and 15 mT. In each image, the sample boundary position was verified as the *y*-value at which the maximum of the sharp induction peak due to the demagnetizing field was observed.

## 3. Results

Figure 2 shows the progression of the magnetic flux density in the unirradiated reference film Y0T at a temperature of 46 K, under two different applied magnetic fields. From the images, we deduce a critical current density $j_c = 1.4 \times 10^{11}$ A·m$^{-2}$ [46]. We shall be interested in the position of the flux front, that is, the distance *h* along the *y*-direction that magnetic flux of density $B_t$ has progressed into the sample, as a function of the lateral coordinate *x*.

Figure 3 compares the penetration of magnetic flux from the long edges of film Y0T, defined by the chemical (left-hand panels (a,c,e)) and ion etching procedures (right-hand panels (b,d,f)), respectively, at a constant applied magnetic field of $B = 25.4$ mT. In both cases, the front is defined by $B_t = 3.7$ mT. Panels (c) and (d) display the higher-order spatial correlation functions $C_q$ (Equation (4)), for *q* values from 2 to 12, evaluated for the flux fronts progressing from the different edges. Here, and throughout the manuscript, the spatial correlation functions were obtained by averaging over the 45 different configurations of the progressing front, one for each value of the applied magnetic field, for a total length of 45 mm. The averaged correlation functions were normalized by the Gaussian factors $R_q^G \equiv C_q^G(x,0)/C_2^G(x,0)$ [17,36], where $C_q^G(x,0)$ terms are the saturation values of $C_q$ that would be obtained for an interface with a Gaussian probability density function of the local displacements [28]. The results show clear power-law scaling, which terminated at a saturation length of $l_{sat} \approx 30$ μm. The power-law scaling also broke down for the very lowest distances (<2 μm), at which the correlations were drowned in the noise. Measurements on different sub-sections of the flux front yielded somewhat

different power laws; the results illustrate those most frequently observed, which also corresponded to the average behaviour.

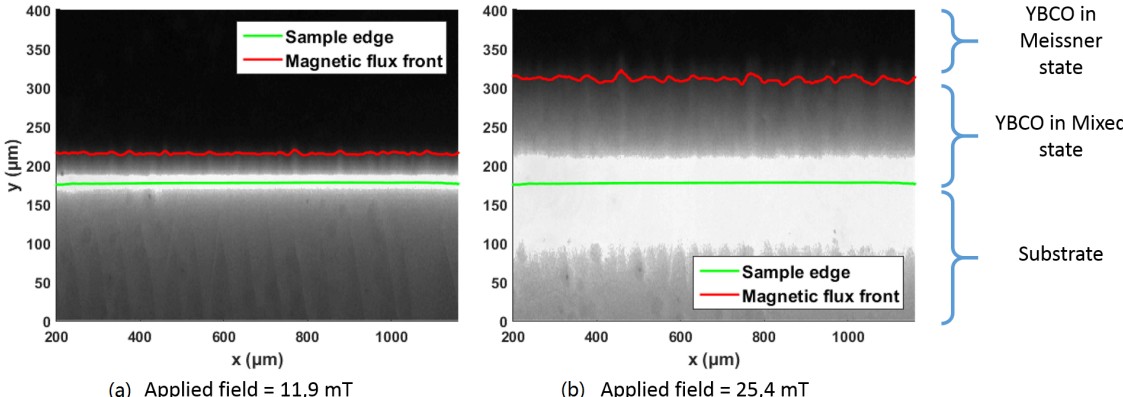

(a) Applied field = 11,9 mT　　　　　　　　(b) Applied field = 25,4 mT

**Figure 2.** Magneto-optical images of magnetic flux penetration in sample Y0T (unirradiated reference film at a temperature of 46 K). Panels (**a**) and (**b**) show the progression of the magnetic flux front at magnetic fields $\mu_0 H_{of}$ 11.9 and 25.4 mT, respectively. The green line depicts the YBa$_2$Cu$_3$O$_{7-\delta}$ (YBCO) film boundary, while the red line is the position of the magnetic flux front, numerically determined at a threshold of 7.5 mT.

We observe, for the flux front progressing from the chemically etched boundary, a clear fanning of the higher-order spatial correlation functions in the regime of small distances below the crossover length $l_\times = 12$ μm. This is indicative of multiscaling. Between $l_\times$ and $l_{sat}$, all the curves are parallel, that is, all moments $C_q$ follow the same power law in $x$. Multiscaling is absent for larger threshold values; see Figure 4a. As for the flux front spreading from the edge defined by ionic etching, the fanning of the $C_q(x,0)$ curves occurs below $l_\times = 8$ μm, and is much less marked. The roughness exponent $\alpha$, or the Hurst exponent $H_q$, retrieved from the data, is plotted as function of the order $q$ in panels (e) and (f). While the flux front progressing from the chemically etched boundary shows an exponent that clearly decreases as a function of the order $q$, from a value of $H_q \approx 0.45$ to $H_q \approx 0.2$, the front at the ion-etched boundary shows a constant roughness exponent of $\alpha \approx 0.35$. For all but the lowest threshold value, the results are independent of $B_t$; see Figure 4b. Thus, the results on the as-grown YBa$_2$Cu$_3$O$_{7-\delta}$ film clearly reveal the effect of different treatments of the sample boundary and different degrees of boundary disorder on the roughening of the flux front.

Figure 5 presents the magnetic flux penetration into the three heavy-ion-irradiated films, in each case, from the boundary defined by the chemical etching procedure. As above, multiscaling of the higher-order moments of the spatial correlation function (Equation (4)) is observed. In the case of the two films with the lowest amount of disorder, that is, Y2T irradiated with 1 GeV Pb ions ($n = 1.2 \times 10^{11}$ cm$^{-2}$) and Y10T irradiated with 130 MeV Pb ions ($n = 5 \times 10^{11}$ cm$^{-2}$), the small-distance behaviour of $C_q(x,0)$, as well as the $q$-dependence of the Hurst exponent $H_q$ and the values of $l_\times$ and $l_{sat}$, are all very similar to that found in the unirradiated film. This indicates that the disorder in the vortex ensemble and the subsequent roughening of the flux front was, principally, introduced by the features of the film boundary. In the film exposed to 1 GeV Pb ions ($n = 3 \times 10^{11}$ cm$^{-2}$) and containing columnar defects with a dose-equivalent field of $B_\phi = 6$ T, however, the fanning out of the $C_q$ curves at small distances is much more marked. The saturation length $l_{sat} \approx 20$ μm is somewhat smaller, no clear crossover regime is observed, and the drop of $H_q$ from 0.45 to 0, as $q$ increases, is more rapid. We note that the critical current density $j_c(46 \text{ K}) = (5.4 \pm 0.3) \times 10^{10}$ A·m$^{-2}$ of this film is much smaller, presumably because of stress induced by the heavy ions implanted in the LAO substrate.

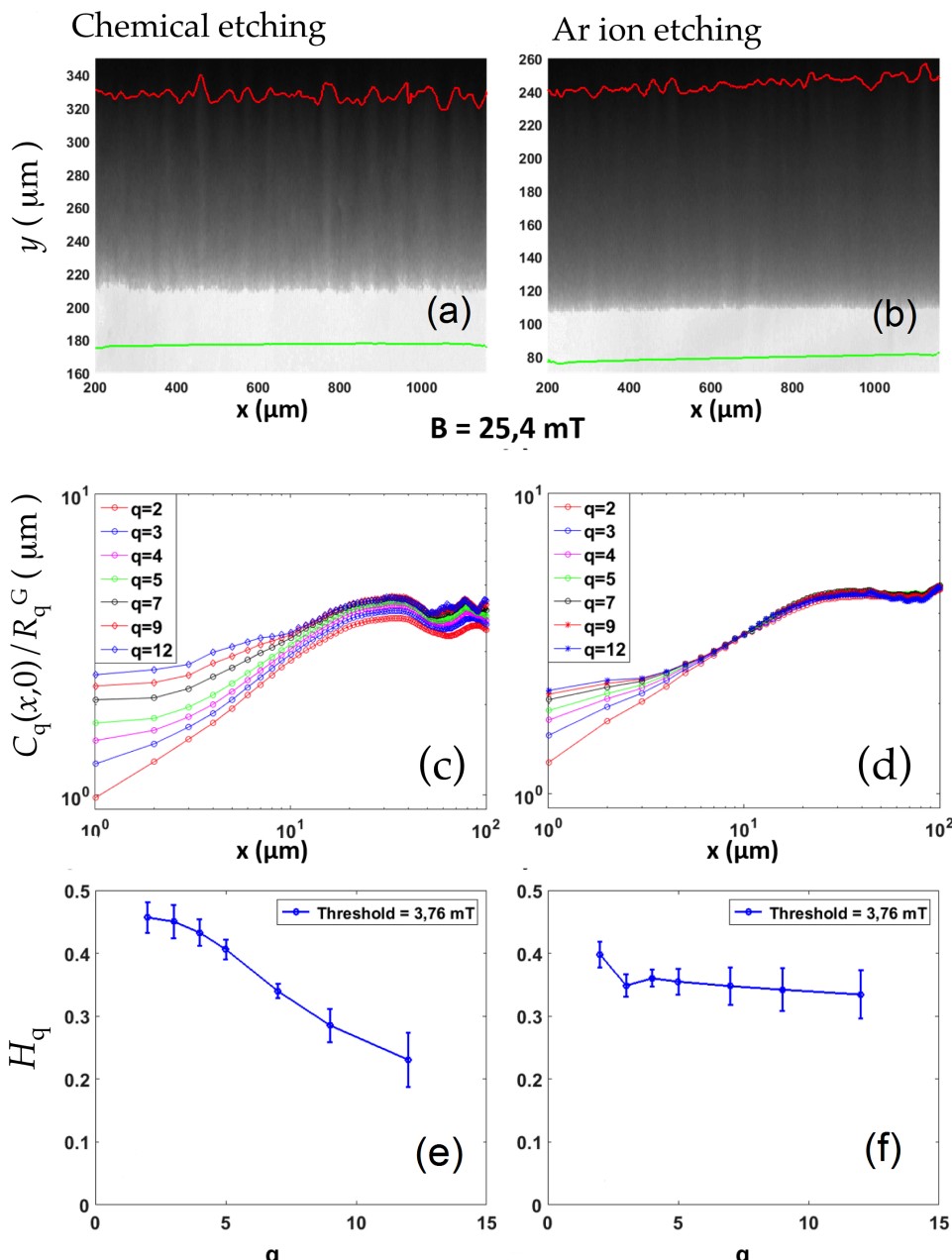

**Figure 3.** The effect of chemical versus Ar ion etching on magnetic flux penetration into $YBa_2Cu_3O_{7-\delta}$ thin-film Y0T. (**a**,**b**) The magneto-optical images of flux penetration from, respectively, the long edge defined by the chemical and the ionic etching procedures, for an applied field of 25.4 mT. The red line indicates the flux front, determined using an induction threshold of $B_t = 3.76$ mT. The green line depicts the sample edge. (**c**,**d**) Higher-order spatial correlation function $C_q(x, t)$, normalized by the Gaussian ratios $R_q^G$ [17,36] with $q = 2, 3, 4, 5, 7, 9, 12$ for chemical etching (**c**) and ionic etching (**d**). Roughness exponent $\alpha$ as a function of $q$ for chemical (**e**) and ionic (**f**) etching.

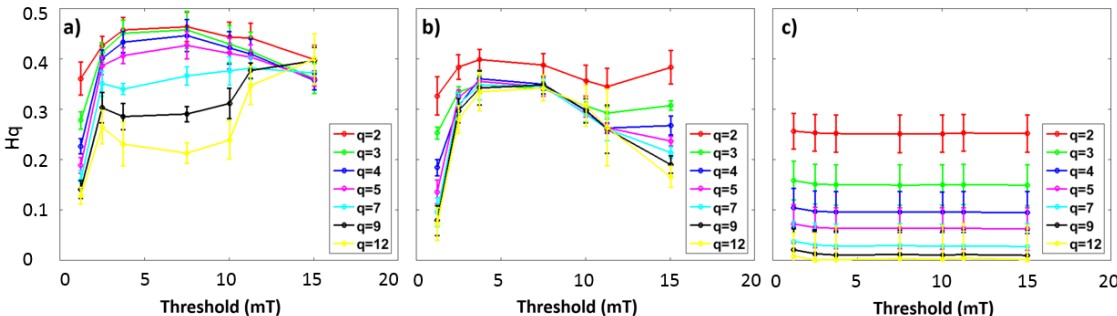

**Figure 4.** Dependence of the generalised Hurst exponent $H_q$ on the value of $B_t$, which is varied from 1.5 to 15 mT, for different orders $q$. The exponents are determined from data equivalent to those in Figure 3, collected on the chemically etched (**a**) and the ion-etched boundaries (**b**) of the non-irradiated reference film Y0T, and on the ion-etched boundary of film Y2T. (**c**) shows $H_q$ as function of $B_t$ for the front emanating form the ion-etched boundary of film Y2T.

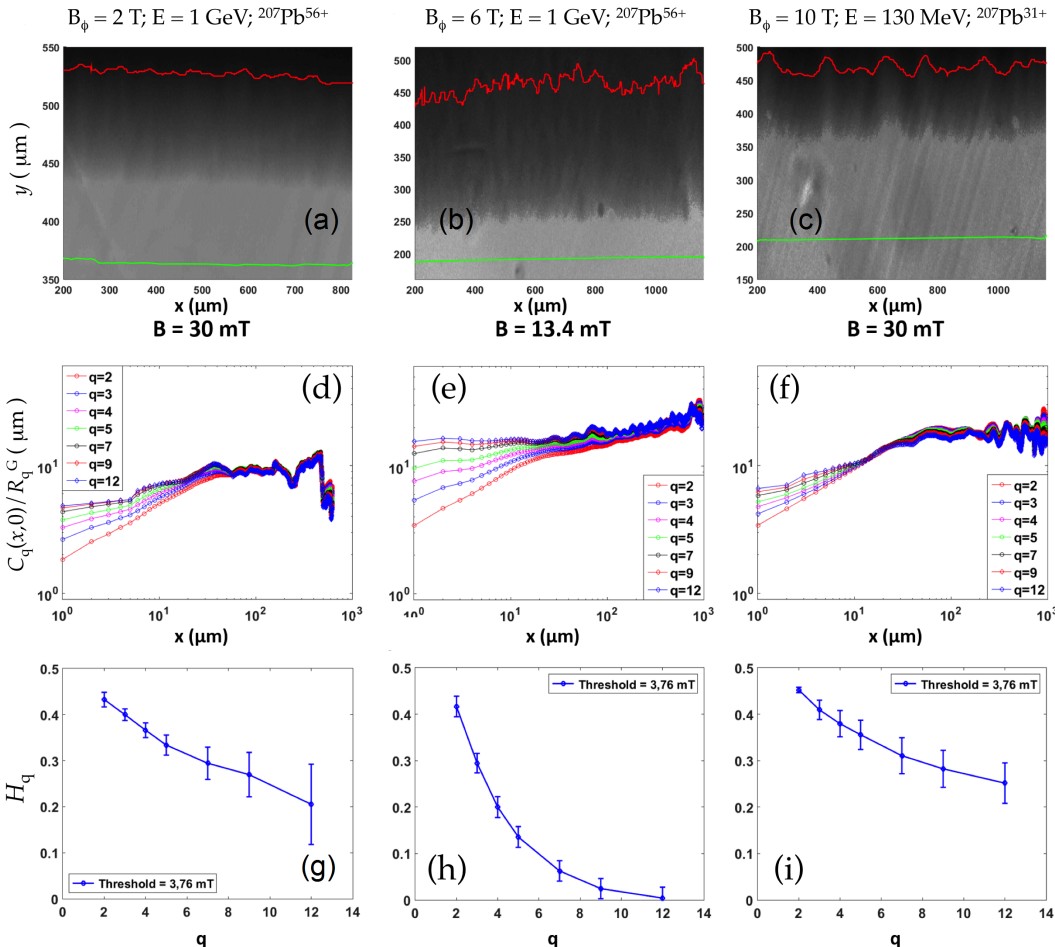

**Figure 5.** Results obtained for the flux fronts progressing from the edges defined by chemical etching on YBa$_2$Cu$_3$O$_{7-\delta}$ films Y2T (irradiated with $1 \times 10^{11}$ cm$^{-2}$ 1 GeV Pb ions, under applied magnetic field of 30 mT; left-hand panels), Y6T (irradiated with $3 \times 10^{11}$ cm$^{-2}$ 1 GeV Pb ions, under applied magnetic field of 13.4 mT; central panels), and Y10T (irradiated with $5 \times 10^{11}$ cm$^{-2}$ 130 MeV Pb ions, under applied magnetic field of 30 mT; right-hand panels). (**a–c**) Magneto-optical images of flux penetration. (**d–f**) Higher-order moments of the spatial correlation function $C_q(x, t)$ (Equation (4)), normalized by the Gaussian ratios $R_q^G$ [17,36]. (**g–i**) Generalized Hurst exponent $\alpha$ as a function of the order $q$.

In the case of flux penetration from the Ar ion-etched boundaries (Figure 6), irradiation with 130 MeV Pb ions ($n = 5 \times 10^{11}$ cm$^{-2}$) resulted in very weak multiscaling and a modest $H_q$ dependence on $q$. Irradiation with 1 GeV ions caused $C_q$ at small distances to fan out in a marked manner. As in the previous case of flux penetration from the chemically etched boundary into film Y6T subjected to 1 GeV Pb ions ($n = 3 \times 10^{11}$ cm$^{-2}$), the Hurst exponent $H_q$ is rapidly suppressed from 0.4 to 0 as the order $q$ increases. This suggests that the irradiation with 130 MeV ions introduced disorder into the vortex system that was comparable to that present in the pristine film and to that introduced by vortex penetration through the ion-etched boundary. On the other hand, the disorder introduced in the vortex ensemble by the introduction of columnar defects by GeV ion irradiation dominated over that induced by native (growth) defects in reactive thermal evaporation-deposited YBa$_2$Cu$_3$O$_{7-\delta}$. The saturation lengths again decreased as the disorder level increased.

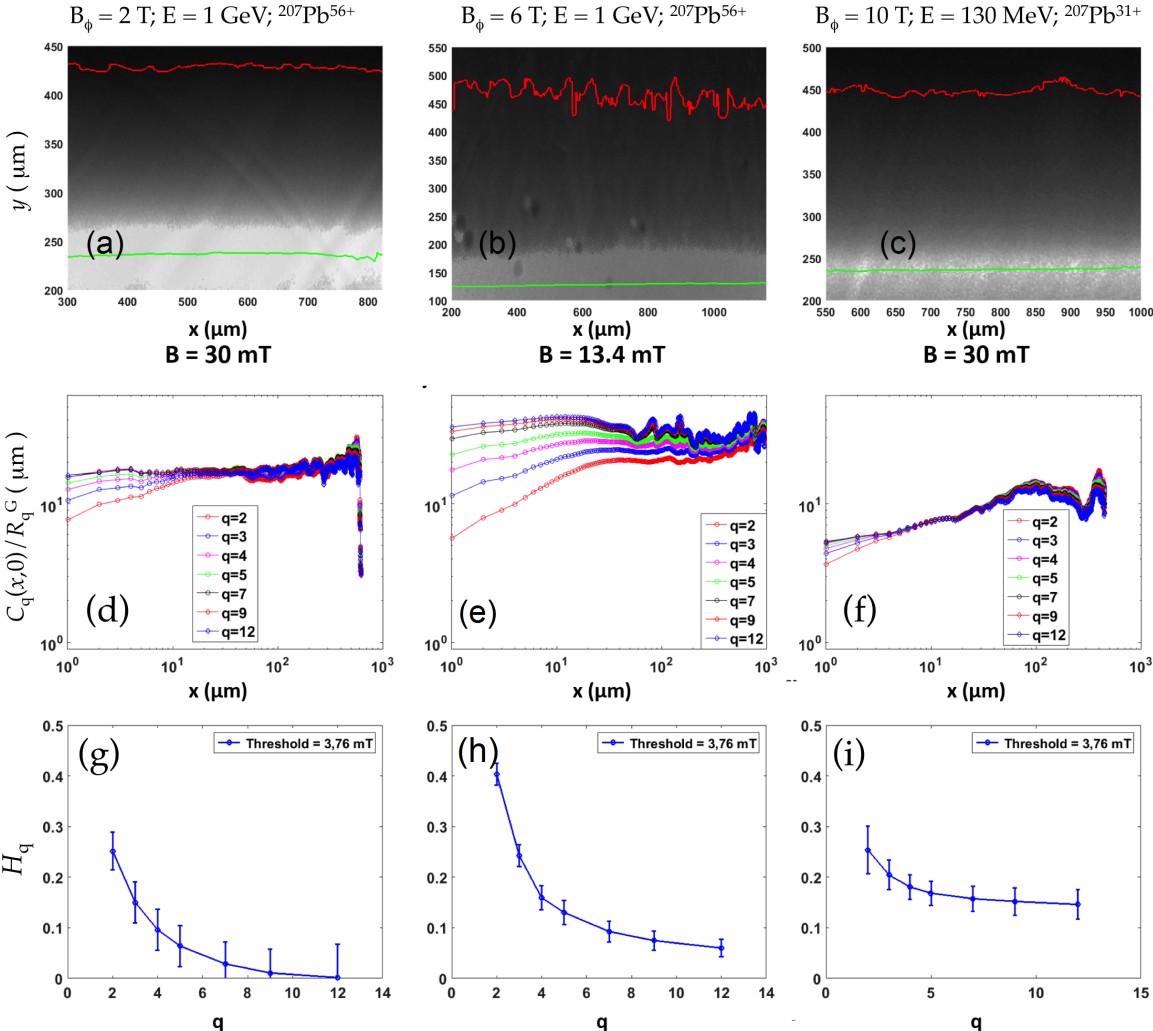

**Figure 6.** Results obtained for the flux fronts progressing from the edges defined by Ar ion etching on YBa$_2$Cu$_3$O$_{7-\delta}$ films Y2T (irradiated with $1 \times 10^{11}$ cm$^{-2}$ 1 GeV Pb ions; $H = 30$ mT; left-hand panels), Y6T (irradiated with $3 \times 10^{11}$ cm$^{-2}$ 1 GeV Pb ions; $H = 13.4$ mT; central panels), and Y10T ($5 \times 10^{11}$ cm$^{-2}$ 130 MeV Pb ions; $H = 30$ mT; right-hand panels). (**a–c**) Magneto-optical images of flux penetration at 46 K. (**d–f**) Higher-order moments of the spatial correlation function $C_q(x, t)$ (Equation (4)), normalized by the Gaussian ratios $R_q^G$[17,36]. (**g–i**) Generalized Hurst exponent $H_q$ as a function of the order $q$.

We now turn to the analysis of the "temporal" correlations, $C_q(0, t)$. Figure 7a–d shows the $q$-th moments $C_q(0, t)$ for flux fronts advancing from the chemically etched boundaries of the four studied films. Weak multiscaling is observed for the three films (Y0T, Y2T and Y10T) with the weakest disorder. The values of the growth exponent $\beta$ obtained from a power-law fit to $C_q(0, t)$ lie between 0.2 (film Y2T) and 0.6. The strongly disordered film Y6T has an intermediate value of $\beta \approx 0.4$. Similar results were found for the flux fronts emanating from the ion-etched boundary, as shown in Figure 8.

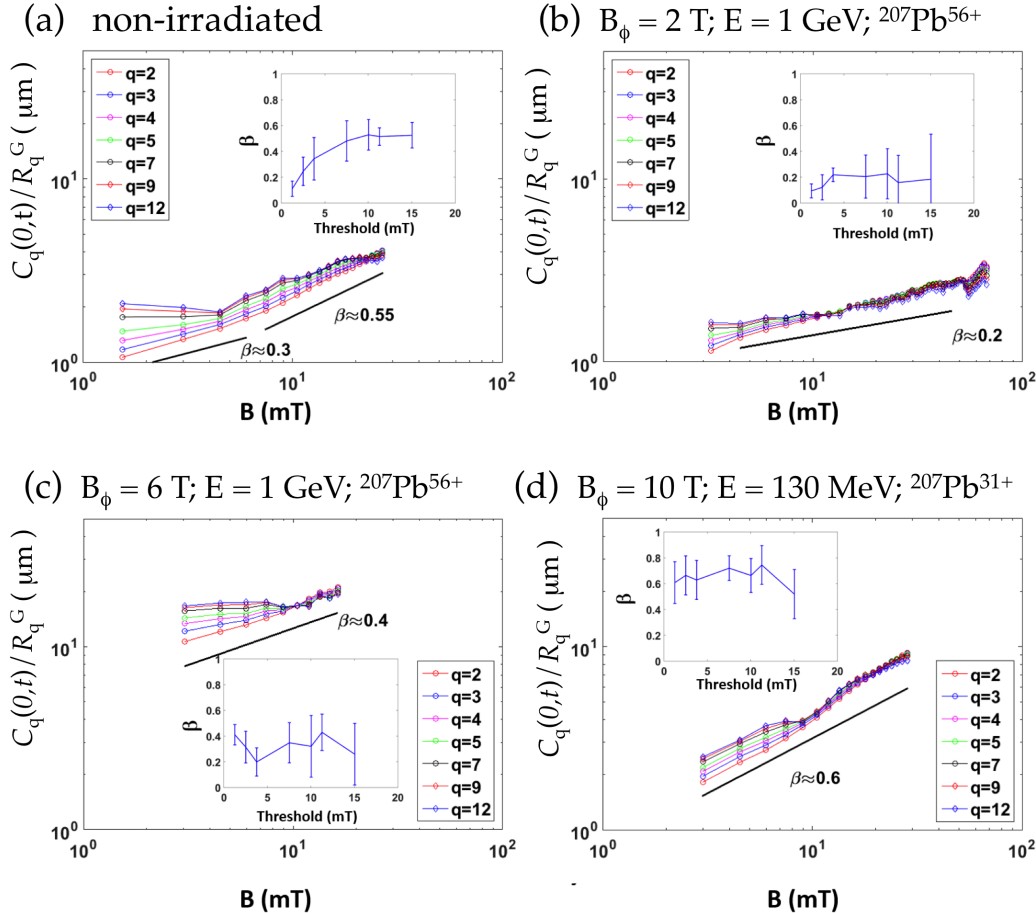

**Figure 7.** Main panels: "Temporal" correlation functions $C_q(0, t)$ describing the growth of flux front features as the magnetic flux advances from the chemically etched boundaries of films Y0T (**a**), Y2T (**b**), Y6T (**c**), and Y10T (**d**). The insets show the growth exponent $\beta$, as extracted from the lowest order $C_2(0, t)$, as function of $B_t$.

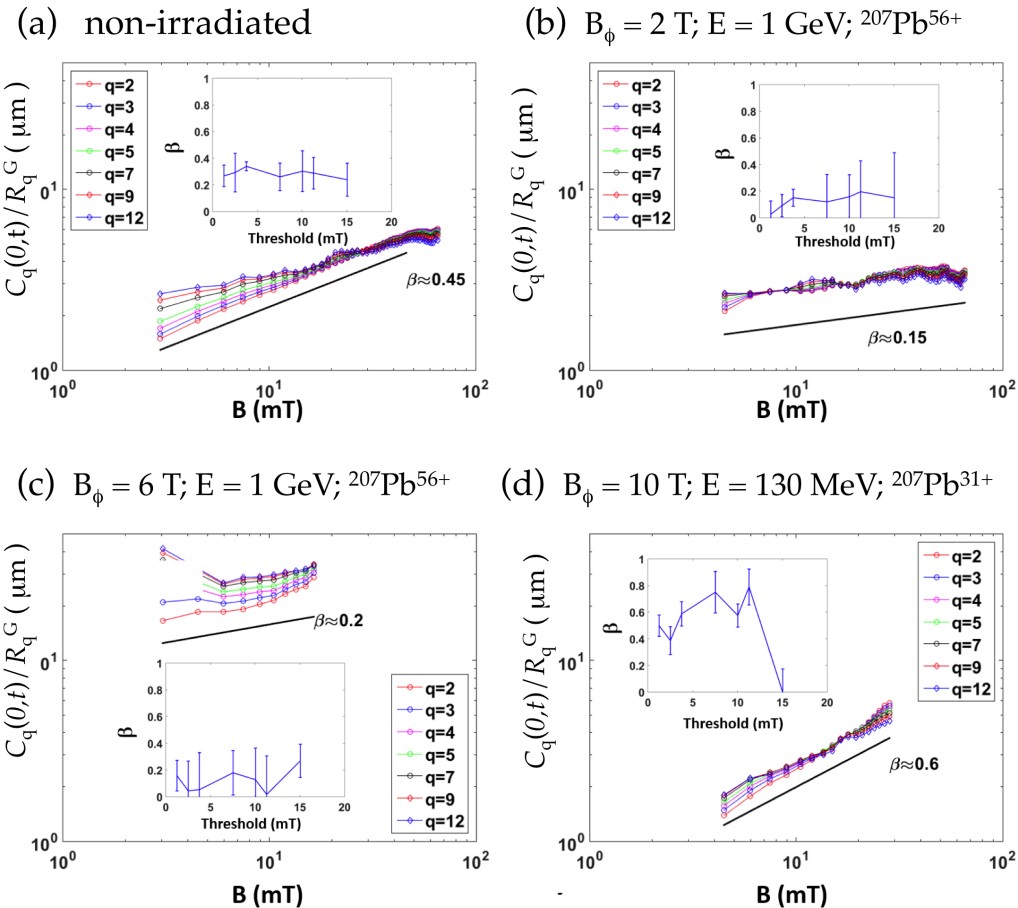

**Figure 8.** Main panels: "Temporal" correlation functions $C_q(0,t)$ describing the growth of flux front features as the magnetic flux advances from the ion-etched boundaries of films Y0T (**a**), Y2T (**b**), Y6T (**c**), and Y10T (**d**). The insets show the growth exponent $\beta$, as extracted from the lowest-order $C_2(0,t)$, as a function of $B_t$.

## 4. Discussion

The results show a clear distinction between the flux fronts progressing from the chemically etched boundaries of the YBa$_2$Cu$_3$O$_{7-\delta}$ films and from the ion-etched boundaries. In the former case, one consistently finds a roughness exponent $\alpha \approx 0.45$ and a growth exponent $\beta \approx 0.3$ that are very similar to those found in the iron-based superconductor Ba(Fe$_{0.925}$Co$_{0.075}$)$_2$As$_2$ [28], and that are consistent with the predictions of the KPZ model. The higher-order moments of the spatial correlation function show three different regimes: at small distances $x$, there is multiscaling, with a clear fanning of the $C_q(x,0)$ curves as the order $q$ increases. The generalized Hurst exponent decreases from $H_q \approx 0.45$ to $H_q \approx 0.2$. At intermediate distances, $l_{sat} > x > l_\times$, the $C_q(x,0)$ curves are parallel on a log–log scale, where $\alpha \approx 0.45 \approx \alpha_{KPZ}$. For $x > l_{sat} \approx 30$ µm, saturation of the correlation functions occurs. The scaling behaviour is relatively insensitive to the introduction of additional bulk disorder by heavy-ion irradiation; only at a very large fluence of 1 GeV Pb ions ($n = 3 \times 10^{11}$ cm$^{-2}$), implicating substantial damage to the film and the substrate, does the qualitative behaviour of $C_q(x,0)$ change, producing an increased length regime over which multiscaling occurs, a vanishing of the intermediate regime, and a smaller saturation length.

Flux fronts progressing from the Ar ion-etched boundaries of as-grown films are characterised by a much-reduced low-distance regime of multiscaling and by a nearly constant $\alpha \approx \beta \approx 0.35$–$0.45$, which is more in agreement with the Oslo model [37]. The introduction of bulk disorder through ion irradiation had a more pronounced effect. If exposure to 130 MeV ions resulted in modest

multiscaling, irradiation with 1 GeV Pb ions ($n = 1 \times 10^{11}$ cm$^{-2}$) led to marked multiscaling; $H_q$ rapidly dropped from 0.3 to 0. This behaviour was also followed for the film exposed to 1 GeV Pb ions ($n = 3 \times 10^{11}$ cm$^{-2}$). Therefore, bulk disorder introduced by irradiation with GeV Pb ions dominated over edge disorder.

For small levels of bulk disorder, the statistical properties of the flux fronts were correlated with the different treatment of the edges from which they evolved, rather than with bulk pinning. Indeed, the pinning "landscapes" and the resulting critical current densities are the same in Figures 5 and 6—the figures concern the same YBa$_2$Cu$_3$O$_{7-\delta}$ films—yet the flux fronts behave differently. This underscores that the roughening of the flux fronts was, for films Y0T, Y2T, and Y10T, essentially determined by edge defects introduced by the chemical etching procedure. We note also that the growth exponent is, in all cases, independent of the edge treatment. This shows that bulk pinning similarly influenced the roughening of the flux fronts induced by the film boundaries. The increasing importance of bulk disorder was revealed by a rapid drop of $H_q$ to 0 as the order $q$ increased.

The general aspect of the scaling behaviour of the correlation functions, with three length regimes, $x < l_\times$, $l_\times < x < l_{sat}$, and $x > l_{sat}$, points to the importance of the occurrence of rare events [23]. In our experiments, these would correspond to the presence of particularly pronounced defects at the superconducting film boundary, responsible for preferential flux penetration [32]. The question of whether these are linked to avalanche-like flux penetration at a low temperature [32,34,35,47–51] remains to be established. Simulations in [23] reveal that the low-distance regime ($x < l_\times$) corresponds to multi-affine behavior, arising from the superposition of growth fronts initiated by different rare events, while the regime of intermediate lengths $l_\times < x < l_{sat}$ signals the return to a self-affine front determined by the progression of vortices through bulk disorder. We confirm these ideas by isolating a particularly marked defect on the chemically etched boundary of film Y2T (Figure 9). Taking the effect of the defect on the flux front into account results in multiscaling; ignoring it lessens multiscaling substantially. We surmise that the crossover length scale $l_\times$ identified above is directly linked to the lateral size of the rare event.

We now turn to the effect of bulk disorder introduced by the different irradiations. Inspection of the TEM images (Figure 10) reveals volumes of approximately 10% and 23% occupied by amorphous defects in films Y2T and Y6T, respectively. In a YBa$_2$Cu$_3$O$_{7-\delta}$ thin film irradiated with 130 MeV $^{207}$Pb$^{31+}$ ($n = 1 \times 10^{12}$ cm$^{-2}$), the defect coverage was 29% of the sample. From this, we deduce coverage of at least 15% for film Y10T ($B_\phi = 10$ T). In spite of this, film Y10T had a higher $T_c$ and presented a weaker disorder potential than Y2T ($B_\phi = 2$ T). This is because, upon irradiation with 1 GeV ions, the regions surrounding the latent tracks are damaged by secondary electrons emitted during ion transit [52]. In film Y6T, 23% of the superconducting material was rendered amorphous by the irradiation. The defect landscape is characterised by a large number of overlapping defects and ensuing irregularly shaped amorphous regions. We surmise that these groups of defects give rise to rare events that permit easy flux penetration in the film bulk and, thereby, front roughening and multiscaling. The presence of some multiscaling in the flux front emanating from the ion-etched edge of the Y2T film suggests that such rare defect groups are already present at lower ion fluences.

The roughness of the investigated flux fronts is important in microscopic superconducting devices, such as antennas, resonators, and filters, especially if these are subjected to external magnetic fields (such as is the case, for example, in magnetic resonance imaging). The flux noise as well as the quality factor of such devices is deteriorated by flux motion, and more notably, by vortex transit through the device. The latter is notably facilitated by flux front roughening and, especially, by the occurrence of rare defect configurations (events). Our work therefore not only identifies the origin of different scaling behaviour of roughened flux fronts, but indicates the pitfalls to be avoided during micro-patterning of superconducting devices.

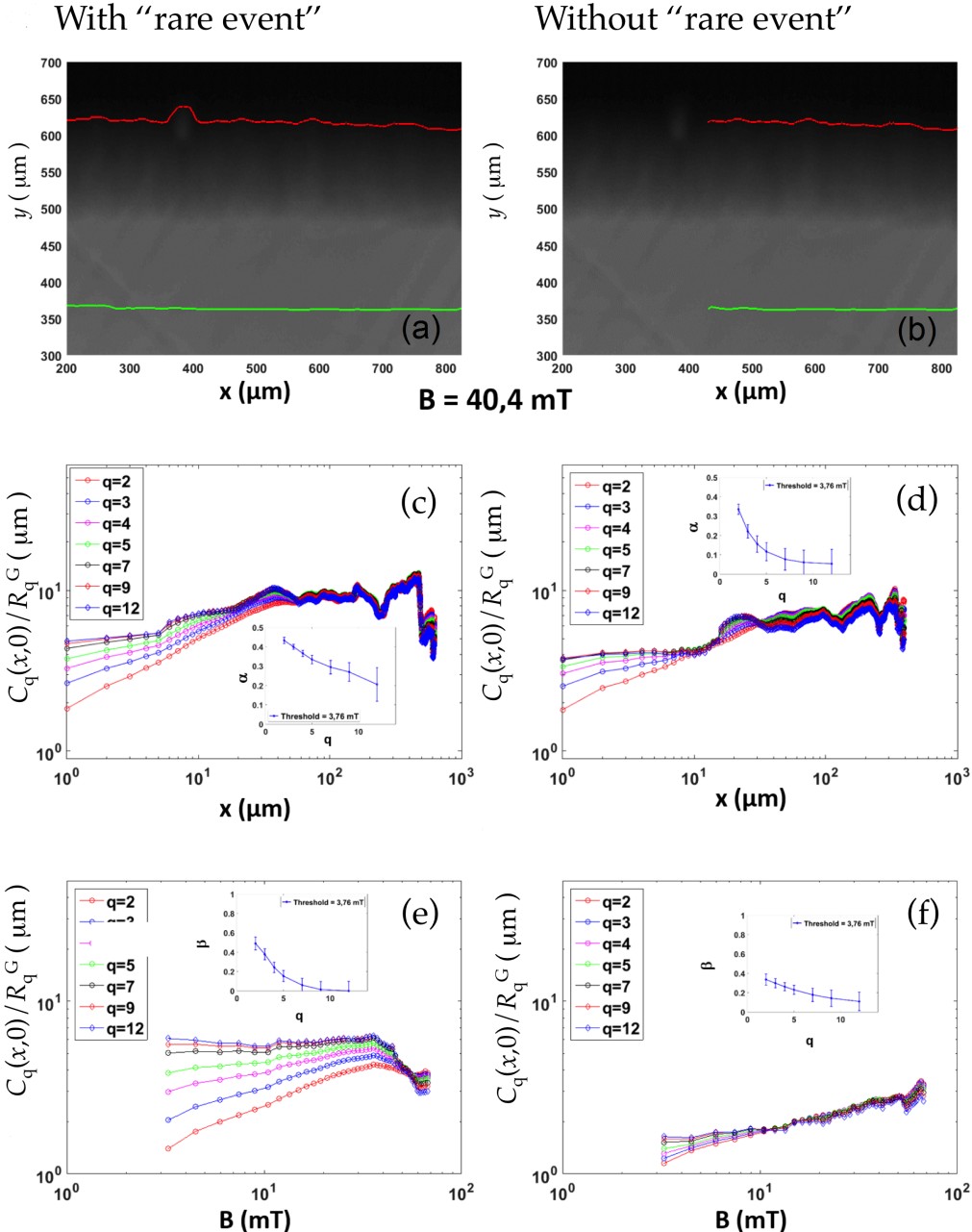

**Figure 9.** Flux penetration from the chemically etched boundary of film Y2T, in a section that includes a particularly marked defect or rare event. The latter causes the bulge in the left-hand side of the flux front (**a**,**b**). Taking the rare event into account results in pronounced multiscaling of the spatial correlation function $C_q(x, 0)$, while ignoring it does not (**c**,**d**). The panels (**e**,**f**) show the corresponding growth exponents. The resulting Hurst exponents are depicted in the respective insets.

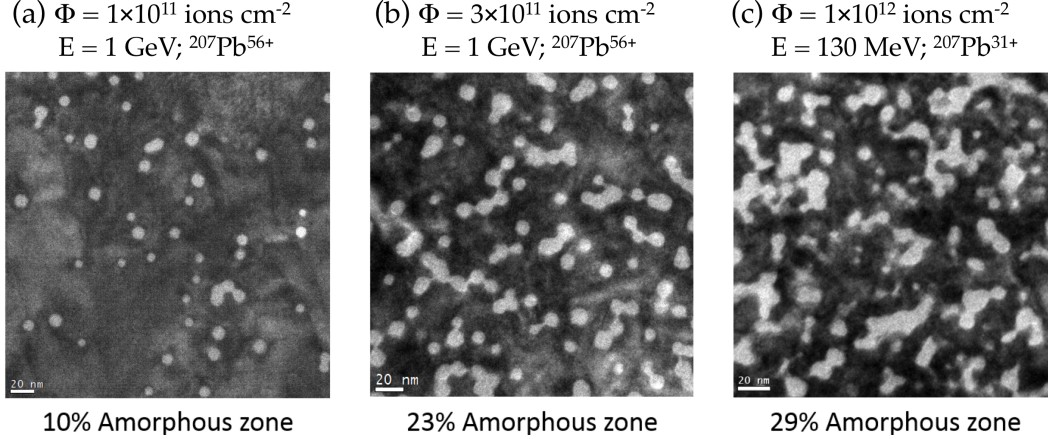

**Figure 10.** Planar view transmission electron microscopy (TEM) images of thin film irradiated with $1 \times 10^{11}$ cm$^{-2}$ (**a**) and $3 \times 10^{11}$ cm$^{-2}$ (**b**) 1 GeV Pb ions, and $10 \times 10^{11}$ cm$^{-2}$ 130 MeV Pb ions (**c**).

## 5. Summary and Conclusions

We have studied the roughening of magnetic flux fronts penetrating into strongly pinning YBa$_2$Cu$_3$O$_{7-\delta}$ superconducting films with different degrees of edgef and bulk disorder. Weak edge disorder was implemented through Ar ion etching the film boundaries, while strong disorder resulted from optical lithography and chemical etching. Increasing the strengths of bulk disorder, corresponding to the as-grown films; irradiation with 130 MeV Pb ions ($n = 5 \times 10^{11}$ cm$^{-2}$); and irradiation with 1 GeV Pb ions ($n = 1 \times 10^{11}$ cm$^{-2}$ and $n = 3 \times 10^{11}$ cm$^{-2}$) were also investigated. Strong disorder in YBa$_2$Cu$_3$O$_{7-\delta}$ superconducting films, be it edge or bulk disorder, was characterised by the occurrence of "rare events" or particularly pronounced defects or defect arrangements, the influence of which led to multi-affine flux fronts and multiscaling. Strong edge disorder yielded a generalized Hurst exponent $H_q$ that decreased from the KPZ value 0.5, and saturated at $H_q \sim 0.2$. This behaviour was very similar to that previously found in superconducting Ba(Fe$_{0.93}$Co$_{0.07}$)$_2$As$_2$ single crystals [28]. Weak bulk disorder resulted in KPZ model- or Oslo model-like behaviour [37], while strong bulk disorder yielded multiscaling, the Hurst exponent rapidly decreasing to 0 as $q$ increased.

**Acknowledgments:** This work was supported by the Region Ile-de-France in the framework of the "Domaine d'Intérêt Majeur", "Nano-K", the French National Research Agency (Agence Nationale de la Recherche) grant ANR-14-CE17-0003 "SupraSense", and the Triangle de la Physique project No. 2013-1019T "Super-IRM".

**Author Contributions:** C.J.v.d.B. conceived and designed the experiments, analysed part of the results, suggested their interpretation, and wrote the paper; M.G. performed the magneto-optical optical observations, analysed the data, and wrote the paper; I.J. performed all the Transmission Electron Microscopy (TEM) work, including sample preparation and all observations; D.G.-P. analysed the TEM data; J.B. contributed reagents and materials, performed the etching procedures, and participated in the interpretation of the data; M.P.-Q. oversaw the magneto-optical optical observations and data analysis.

**Conflicts of Interest:** The authors declare no conflict of interest.

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
