# Peer review of "Edge Contamination, Bulk Disorder, Flux Front Roughening, and Multiscaling in Type II Superconducting Thin Films"

_condensedmatter, doi:10.3390/condmat2030027_

Reviewer 1 Report

I have read this paper with great interest. I commend the authors for high quality of their work, detailed experiments, and clear and substantiated discussion. In principle, the article is very well written, and the topic and results merit publication in CondMat.

Reading through the paper, I only had minor remarks considering references, where the work about flux front propagation and scaling laws in  PRB 66, 174507 (2002) should be mentioned, as well as the works by Wilson Ortiz and Alejandro Silhanek about guiding of flux avalanches by pinning lattices. Among very recent works, I found arxiv:1706.00628 (to appear in NatCommun) as a beautiful experiment on vortex penetration, motion, and extreme velocity, that is clearly relevant to the present paper (and increases its contemporary value).

In any case, I recommend the paper for publication.

Author Response

We thank Referee 2 for his thorough reading and very positive assessment of our work. We also thank her / him for reminding us of the work by Zapperi, Moreira, et al.. While we were aware of this, the work by these authors describes the progression of the mean front position under a variety of different experimental conditions, only one of which (condition D in ref. [2] below) is directly relevant to our experiment. This condition is only very briefly discussed in Ref [2]. Also, [1] and [2] concern the mean progression of the front, rather than its roughening, which occupies only a minor part of those references (no explicit calculations of the growth and roughness exponents for the roughness are provided for example). Nevertheless, the work in Refs. [1] and [2] below is relevant to the work and quite elegant. In the revised version of the manuscript, we therefore make reference to it in the introductory section, Refs. [13] and [14] (new numbering).

We also thank the Referee for pointing out Ref. [3] (in the numbering below), that strongly questions the direct role of geometrical defects at the superconductor boundary in initiating flux avalanches. While such avalanches may be linked to edge disorder in our YBa2Cu3O7 films, to establish this requires further study. We therefore change the phrase on lines 234-236, which now reads “…, responsible for preferential flux penetration [50].  The question of whether these are linked to avalanche-like flux penetration at low temperature [33,37,47–52] remains to be established”, which also allows for a reference to [4] and [5] below.

[1] Zapperi S, Moreira A A, and Andrade Jr. J S  2001 Flux Front Penetration in disordered Superconductors Phys. Rev. Lett. 86, 3622--3625
[2] Moreira A A, Andrade Jr. J S, Filho J M, and Zapperi S, Boundary effects on flux penetration in disordered superconductors Phys. Rev. B 66 174507
[3] Brisbois J, Adami O-A, Avila J I, Motta M, Ortiz W A, Nguyen N D, P. Vanderbemden P, Vanderheyden B, Kramer R B G, and Silhanek A V 2016 Magnetic flux penetration in Nb superconducting films with lithographically defined microindentations Phys. Rev. B 93, 054521

Reviewer 2 Report

This is a very interesting experimental work, which addresses the important issue of the role of different types of disorder on the scaling properties of magnetic flux front in type II superconductors. These flux fronts are studied as fluctuating elastic interfaces in a disordered energy landscape, and as such they connect to established theoretical models and predictions in the generic framework of disordered elastic systems. From a fundamental point of view, such experimental studies comparing different types of disorder are scarce and thus very precious. The results are moreover relevant for applications using type-II superconductors.

In my opinion this paper should be published in Condensed Matter. However, before doing so, there are a few issues that I believe should be addressed by the authors, but which would correspond at most to minor modifications.

1) My first main issue is that, not being familiar at all with this specific type of experiment, I did not found the geometry of the experimental setup very clear. A schematic picture would be very helpful to understand it better, with more comments in the model section. In particular, the connection between the vortices and the measured magnetic flux fronts could be clarified further (are the latter a single vortex flux front?).

2) My second main issue concerns the statistics used for the different analyses. In Ref. 15 by Guyonnet et al. the same analysis was performed (on ferroelectric domain walls) in order to detect the possible presence of strong pinning centers. But for interfaces that did not had any of them, there was not a unique value for the roughness exponent for individual domain walls, but rather a rather large distribution. This was shown to also be the case for numerical interfaces when they are too small (512 pixels long) for a very precise determination of the roughness power-law behaviour. In the present study, how many different interfaces were used to measure their scaling exponents, and what were their total length? At least a short comment on that could be useful to clarify what is done to measure the exponents, and to establish their robustness.

3) In the definition of the model around Eq. (1) I found some imprecisions. First, some references at the first occurrence of the DPD and KPZ models would be welcome. Secondly, the stochastic noise eta(x,h;t) is assumed to be Gaussian, but the two-point correlator that is given just after depends on eta(x,t) instead of eta(x,h,t), so that it is a bit confusing when first reading this paragraph.

Author Response

We thank Referee 1 for her / his careful reading of the manuscript, positive appraisal, and very useful questions and remarks.

In response to Referee 1’s request, we include a supplementary figure sketching the experimental set-up, with a scheme illustrating magnetic flux penetration into the superconducting thin film strips, and how large numbers of vortices contribute to the total magnetic flux. In this scheme, the flux front is schematically illustrated. The Figure is combined with large-area imaging of the four superconducting YBa2Cu3O7 samples, as requested by Referee 3. The different panels of the new Figure 1 are referenced to in relevant parts of the introductory text - we hope that the Referee will find the added material sufficiently clear to better grasp the manner in which the experiment is set up.

The presented data concern interfaces on four different samples outlined in the text. For each sample, the growth exponent was extracted from the scaling behaviour of the correlation functions (2), (5), averaged over the “time” variable t - actually the magnetic field value. Thus, in accordance with the experimental details on line 110, each progressing interface is effectively sampled 45 times at different locations in the same film. The graphs rendering the correlation functions (Figures 3, 4, and 5) thus average over 45 configurations of the same, progressing interface.  We have also performed analyses of sub-sections of each interface, and find, as Guyonnet did, a rather wide distribution of roughness exponents. It is now specified in the text that the illustrated exponents correspond to the mean (which equals, in this work, the mode) : 
The presented data concern interfaces on four different samples outlined in the text. For each sample, the growth exponent was extracted from the scaling behaviour of the correlation functions (2), (5), averaged over the “time” variable t - actually the magnetic field value. Thus, in accordance with the experimental details on line 110, each progressing interface is effectively sampled 45 times at different locations in the same film. The graphs rendering the correlation functions (Figures 3, 4, and 5) thus average over 45 configurations of the same, progressing interface.  We have also performed analyses of sub-sections of each interface, and find, as Guyonnet did, a rather wide distribution of roughness exponents. It is now specified in the text that the illustrated exponents correspond to the mean (which equals, in this work, the mode).

Line 169: “ Here, and throughout the manuscript, the spatial correlation functions are obtained trough averaging over the 45 different configurations of the progressing front, one for each value of the applied magnetic field, for a total length of 45 mm.  … “ ;

Line  177: “Measurements on different sub-sections of the flux front yield somewhat different power laws; the results illustrate the most frequently observed, which also corresponds to the average behaviour.”

We have also added two References to the Directed Percolation Depinning model, as well as the original references to the Edwards-Wilkinson and Kardar-Parisi-Zhang models :

[1]  Havlin S, Barabási A-L,  Buldyrev S~V,  Peng C~K, Schwartz M, Stanley H E, and Vicsek T 1991  Anomalous Surface Roughening: Experiment and Models  in  Growth Patterns in Physical Sciences and Biology [Proc. 1991 NATO Advanced Research Workshop, Granada] ed. J.M. Garcia-Ruiz, E. Louis, P. Meakin, and L.M. Sander (Plenum Press, New York, 1993)

[2] Buldyrev S V, Barabási A-L, Havlin S, Caserta F, Stanley H E, and Vicsek T 1992 Anomalous interface roughening in porous media: Experiment and model  Phys. Rev. A 45, R8313

[3]  Edwards S F and Wilkinson D R 1982 The Surface Statistics of a Granular Aggregate Proc. Roy. Soc. London, Ser. A 381, 17

[4] Kardar M, Parisi, G, Zhang, Y-C 1986 Dynamic Scaling of Growing Interfaces  Physical Review Letters 56 889–892

Reviewer 3 Report

referee report MDPI condensed matter 211023 Michel Geahel et al. Edge contamination, bulk disorder, flux front roughening, and multiscaling in type II superconducting thin films

This article deals with a very interesting topic -- the understanding of the flux front roughening in type-II superconducting thin films. The topic is well suited for this journal, and the article is well written and organized, which also applies for all the figures. The reference list given is fairly complete and gives a full picture of this type of research. There are only two minor points to be mentioned.

(1) In order to be able to judge about the quality of the samples investigated, I would have expected a complete overview on the samples with low magnification. It would be interesting to see the discontinuity lines and eventually existing defects.

(2) For which reason is the title of Ref. 36 written in capital letters? Overall, this article is well executed and should be published.

Author Response

We thank the Referee for her / his careful reading of the manuscript and her / his very encouraging comments. In response, we add a new Figure 1, that shows, among others, magneto-optical images with low magnification on all films. The films being much longer than wide, we have not taken images of the short edges. The new Figure 1 combines these images with a scheme of the experimental set-up, and a schematic illustration of flux penetration t-int the superconducting YBa2Cu3O7 strips, as requested by Referee 1. Also, there is no particular reason for the capitalisation of the title in Ref. [36] ([44] in the new numbering) - this has been undone in the revised version.
